# RayE-Sub: Countering Subgraph Degradation via Perfect Reconstruction

## Abstract

Subgraph learning has dominated most practices of improving the expressive power of Message passing neural networks (MPNNs). Existing subgraph discovery policies can be classified into node-based and partition-based, which both achieve impressive performance in most scenarios. Unfortunately, we observe that there exists a subgraph degradation trap in these two mainstream solutions. This means extracted subgraphs fail to achieve better expression. In this work, we start with an intuitive observation and theoretical analysis to explore subgraph degeneration. We then summarize the limitations of these two subgraph strategies from the perspective of reconstruction ability. To this end, we propose perfect reconstruction principle to realize high-quality subgraph extraction. To achieve this, two affiliated questions should be well-addressed. *(i) how to ensure the subgraphs possessing with 'perfect' information? (ii) how to guarantee the 'reconstruction' power of obtained subgraphs?* Firstly, we propose a subgraph partition strategy *Rayleigh-resistance* to extract non-overlap subgraphs by leveraging the graph spectral theory. Secondly, we put forward the Query mechanism to achieve subgraph-level equivariant learning, which guarantees subgraph reconstruction ability. These two parts, *perfect subgraph partition* and *equivariant subgraph learning* are seamlessly unified as a novel *Rayleigh-resistance Equivariant Subgraph learning* architecture (**RayE-Sub**). A series of experiments on both synthetic and real datasets demonstrate that our approach can consistently outperform previous MPNNs architectures.

## 1 Introduction

Message Passing Neural Networks (MPNNs) have been widely applied in graph representation due to their simplicity, intuitiveness and effectiveness. Plenty of MPNN variants have achieved great success in various fields, such as molecular discovery, social network analysis and traffic flow prediction (Yang et al., 2022; Zhang et al., 2022; Zhou et al., 2020). However, it has been proved that these architectures exist limited expressiveness, which are at most as expressive as 1-dimensional Weisfeiler-Lehman (1-WL) test (Xu et al., 2018; Morris et al., 2019). To this end, numerous recent researches have proposed more powerful architectures (Zhang et al., 2023; Miao et al., 2022; Cotta et al., 2021; Frasca et al., 2022; Bevilacqua et al., 2022; Zhao et al., 2021; Kreuzer et al., 2021). Among them, subgraph learning is one of the most predominant solution.

Subgraph representation learning aims to extract a bag of subgraphs from an original graph, and explore more powerful expressive methods based on subgraphs (Frasca et al., 2022). Based on the partition perspective of different subgraph discovery solutions, we can classify subgraph learning into two main research lines. (i) Node-based subgraph discovery policy employs predefined structure to extract subgraphs, wherein every subgraph is centered with a unique node in the graph. The implementations of this category include node-deletion, node-marking, and ego-network subgraph extraction policies (Bevilacqua et al., 2022; Zhao et al., 2021). (ii) Partition-based subgraph discovery policy extracts a bag of non-overlapping subgraphs from original graphs. This category includes high-frequency substructure extraction, nodes clustering and edges dropping (Preti et al., 2023; Miao et al., 2022). Both of them have revealed effectiveness in practices, where the former is with more simplicity, while the latter is more interpretable.

We further reflect on these subgraph learning strategies. However, we observe that both of them suffer the limitations potentially resulting in degeneration. For node-based subgraph learning, it may inevitably generate subgraphs with severe overlaps. This kind of overlaps make the extracted local information during subgraph learning be similar to the aggregated information of node-level MPNNs. And the homogenization of information may eventually lead to the degradation of subgraphs in terms of representation power (Top Panel of Figure 1). For partition-based subgraph discovery policy, researchers usually focus on how to implement effective subgraph partition strategies, but ignore the order information of subgraphs. This results in limited distinguishing ability in many scenarios, such as the failure of discriminating most isomers in chemistry (Bottom Panel of Figure 1).

Given above, we summarize existing limitations of previous subgraph learning strategies into the perspective of reconstruction ability. We observe that both subgraph learning methods are with a common issue, i.e., the obtained subgraphs fail to reconstruct the original graph perfectly. The subgraphs generated from node-based strategy will be redundant for reconstructing because of overlapping information, while the subgraphs extracted from partition-based methods fail to possess the inverse reconstruction ability. Thus, we can summarize that their failures into the limitations of reconstruction ability. Actually, if a set of subgraphs can perform exact reconstruction without any redundancy, we can designate these subgraphs as perfect reconstruction, and then the interpretability and generalization of subgraph learning will be greatly promoted. Therefore, this naturally raises the question that *how to obtain subgraphs with perfect reconstruction ability?*

**Present work.** We provide the answers to above question by introducing two sub-solutions: (i) partitioning graph to obtain non-overlap subgraphs ensures the 'perfect' property, (ii) utilizing subgraph-level equivariance principle guarantees the 'reconstruction' property. Specifically, we propose a partition-based strategy guided by spectral graph theory, and employ the Query mechanism to achieve subgraph-level equivariant learning. Firstly, partition-based strategy is more intuitive and interpretable to reflect the real-world facts, such as finding functional groups in molecules, decoupling subnetworks in social networks, and discovering urban functional patterns in cities (Jin et al., 2020; Zhang et al., 2022; Zhou et al., 2020). The commonality of these scenarios lies in finding the physically meaningful boundary among complex and diverse connections. Spectral theory is with power to draw graphs and find the potential boundary from spectral domain (Spielman, 2019; Kreuzer et al., 2021). Therefore, we propose a spectrum-based subgraph partition strategy *Rayleigh-resistance*. Although a bag of subgraphs obtained by partitioning have appropriate and non-redundant information, they still lack reconstruction ability. Secondly, to significantly boost the reconstruction ability, we utilize equivariance principle to investigate the equivalent relationship among subgraphs, thus guarantee the reconstruction power (Bevilacqua et al., 2022). Specifically, we propose a *Siamese-Query* scheme to implement our equivariant architecture, where a Siamese network processes each subgraph independently with same parameters, and the Query mechanism aggregates all subgraphs with their order information. Altogether, above two parts composed of our **Rayleigh-resistance Equivariant Subgraph learning architecture** (**RayE-Sub**), which achieves the perfect reconstruction of extracted subgraphs.

We provide thorough theoretical analysis and comprehensive empirical evaluation. On the theoretical side, we first prove the subgraph degeneration of node-based subgraph discovery by investigating the relationship between subgraph learning and node-wise MPNN. Then we demonstrate that resistance distance can be an alternative to *Rayleigh quotient*, to characterize the graph stability for discovering subgraph boundary. On the empirical side, we show the limited indistinguishability of existing subgraph learning by two families of examples, and verify the competitive performance of our approach on both synthetic and real datasets.

## 2 RELATED WORK

**Subgraph Learning.** According to different subgraph discovery strategies, subgraph learning can be classified into node-based (Frasca et al., 2022; Bevilacqua et al., 2022; Zhao et al., 2021; Cotta et al., 2021) and partition-based learning (Yang et al., 2022; Jin et al., 2020; Miao et al., 2022) paradigm. ESAN (Bevilacqua et al., 2022) implements an subgraph equivariant learning architecture and achieves better expressiveness by per-layer aggregation across subgraphs. SUN (Frasca et al., 2022) profoundly studies the characteristics of node-based subgraph learning. Further, SUN aligns the permutation group of nodes and subgraphs, and models the symmetry with a smaller single

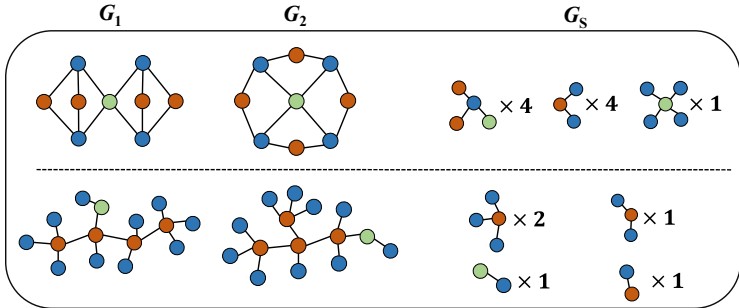

Figure 1: Illustration of subgraph failures. **Top panel:** $m = 2, k = 2$ in Example 1, they will generate the same EGO-based subgraphs. **Bottom panel:** two isomers cannot be distinguished by partition-based methods, 2-Butanol and 2-Methyl-1-Propanol.

permutation group. GSAT (Miao et al., 2022) is a representative partition-based learning method. Guided by the information bottleneck theory, GSAT designs a subgraph extraction strategy with edge deletions based on stochastic attention mechanism. However, in subgraph learning, there is still no uniform architectures to achieve the tradeoff between partition and node-based subgraph discovery.

**Expressive power of MPNNs.** Exploring more expressive learning architectures is the primary goal in graph representation learning. Current researches tend to be divided into three lines, i.e., the MPNN-based methods aligned with WL-Test, transformer-based and solutions derived from novel representation power measures. First, MPNN-based methods improves expressiveness on WL-Test by devising higher-order message-passing (Bodnar et al., 2021; Morris et al., 2019; 2020), position and structure encoding (Chen et al., 2020; Puny et al., 2020; Dwivedi et al., 2021). However, the computational cost for $k > 3$ expressive power in WL-Test should be unacceptable. Second, instead of traditional message passing, transformer like Graphormer computes the soft attention scores for aggregation (Ying et al., 2021). Third, Zhang et al.(Zhang et al., 2023) take a novel perspective, the graph bi-connectivity, as measure of expressiveness and promotes the representation on bi-connectivity aspect. In this work, we inherit the third line and further exploits elegant theoretical paradigms, to find a more interpretable perspective to realize more powerful representation.

## 3 SUBGRAPH LEARNING FROM RECONSTRUCTION PERSPECTIVE

In this section, we first dissect the existence of limited expressive power in both node-based and partition-based approaches, and then summarize the limitations into the limited reconstruction ability.

**Notation.** Let $G = (A, X)$ be an undirected graph with $n$ nodes. The adjacency matrix $A \in \mathbb{R}^{n \times n}$ denotes $G$'s edge set $E$ over its set of $n$ nodes $V$. The feature matrix $X \in \mathbb{R}^{n \times d}$ represents the features of all nodes, where $x(u) \in \mathbb{R}^{1 \times d}$ is the feature of $u$. Let $[n] = 1, ..., n$. $G_S = \{G_S^1, \cdots, G_S^k\}$ represent the subgraph set generated by subgraph discovery policy $\pi(G)$. Each subgraph is $G_S^i = (A_S^i, X_S^i)$ with $V_S^i \subseteq V, E_S^i \subseteq E$, where $1 \leq i \leq k$. We denote $x_i \in \mathbb{R}^{|V_i| \times d}$ as the feature of all nodes in subgraph $G_S^i$, which is different from $x(u)$.

### 3.1 NODE-BASED SUBGRAPH LEARNING

Node-based subgraph discovery manner has become the most popular policy in subgraph learning due to its simplicity and effectiveness (Frasca et al., 2022; Bevilacqua et al., 2022; Zhao et al., 2021). The specific subgraph discovery strategies consist of node-deletion (ND), node-marking (NM), and ego-networks (EGO) policies. Our work is inspired from an observation that two family graphs (Zhang et al., 2023) can be indistinguishable for node-based subgraph learning in Example 1. Specifically, we observe that two subgraph sets generated by these two families of graphs are not distinguishable, as shown in Figure 1.

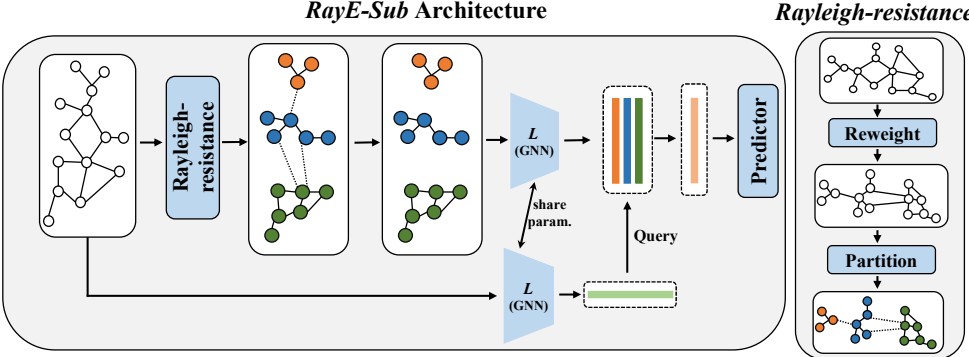

Figure 2: The architecture of *RayE-Sub*. **Left panel:** our *RayE-Sub* is composed of two stages: a partition block *Rayleigh-resistance*, a subgraph-level equivariant module *Siamese-Query*. **Right panel:** the detailed process of *Rayleigh-resistance*.

This observation inspires us to investigate the theoretical interpretation in subgraphs failures. We theoretically prove the existence of this degeneration from the perspective of message passing. Based on (Arora et al., 2016), we derive the Lemma 1 regarding function composition.

**Lemma 1.** *(Function composition (Arora et al., 2016).) MPNNs can repeatedly update each node's embedding by aggregating information from their neighbors. The graph-level embedding $\mathbf{h}_G$ can be obtained by*

$$h_G = \text{POOL} \circ \sigma \circ \mathbf{T}_L \circ \cdots \circ \mathbf{T}_2 \circ \sigma \circ \mathbf{T}_1(A, X) \tag{1}$$

*where $\sigma$ and $\mathbf{T}_1$ are the $l$-th layer update and aggregation operators (matrices),* POOL *is the* READOUT, *$\circ$ denotes layer composition. There exists a global aggregation matrix $\mathbf{T}$, merging from $\{\mathbf{T}_1, ..., \mathbf{T}_l\}$, to make this architecture have equivalent representation:*

$$\mathbf{h}_G = \text{POOL} \circ \sigma \circ \mathbf{T}(A, X) \tag{2}$$

*where each element $\mathbf{T}(i, j)$ indicates the aggregation coefficient from $j$ to $i$.*

We can obtain a profound understanding about message passing mechanism, which is essentially to explore the aggregation coefficient among nodes. If extracted subgraphs fail to limit previous MPNN' aggregation trend, they will not bring better ability of distinguishing (Alsentzer et al., 2020; Yuan et al., 2021). Detailed proof is provided in Appendix C.1.

**Theorem 1.** *(The existence of subgraph degradation.) Let $\mathbf{M}$ be the ground-truth aggregation matrix of graph $G$, and the aggregation matrix conducted by node-level MPNNs be $\mathbf{T}_G$, while the aggregation matrix implemented by node-based subgraph learning be $\mathbf{T}_S$. Then, there exists graph $G$ satisfying $\mathbf{T}_S = \mathbf{T}_G$.*

*Proof.* Node-based subgraph learning generates $n$ subgraph set $G_S = \{G_S^1, G_S^2, \cdots, G_S^n\}$, wherein every subgraph is associated with a unique node. The aggregation process of each subgraph $G_S^i$ obtains a corresponding aggregation matrix $\mathbf{T}_S^i$. For the whole graph, the aggregation matrix of information passing among all nodes is $\mathbf{T}_S = \sum_{i \in [n]} \mathbf{T}_S^i$. Therefore, we turn the question into proving the existence of $\sum_{i \in [n]} \mathbf{T}_S^i = \mathbf{T}_G$. Detailed proof is provided in Appendix C.2.

Theorem 1 indicates the existence of subgraph degradation, which verifies our intuitive observation. Note that the degradation of subgraphs is not universal, but it doesn't alter the fact that the expression ability of subgraph learning is strictly more powerful than 1-dimensional Weisfeiler-Lehman (1-WL). More discussion is provided in Appendix C.3.

## 3.2 PARTITION-BASED SUBGRAPH LEARNING

Partition-based subgraph learning aims to extract a subgraph set $G_S = \{G_S^1, G_S^2, \cdots, G_S^k\}$ which are with no overlaps among any pair of elements, as described in Equation 3 and 4.

$$\mathcal{V} = \mathcal{V}_S^1 \cup \mathcal{V}_S^2 \cup ... \cup \mathcal{V}_S^k \tag{3}$$

$$\mathcal{V}_S^i \cap \mathcal{V}_S^j = \emptyset \ (\forall i, j \in [k], i \neq j) \tag{4}$$

Given that various practices aim to refine and obtain a minimal label-relevant subgraph set (Miao et al., 2022; Wu et al., 2022), Equation 3 is not a compulsory condition. Compared with node-based subgraph strategy, partition-based is more intuitively in real-world tasks, such as finding the functional groups in molecules (Yang et al., 2022).

However, most previous works (Miao et al., 2022; Wu et al., 2020; Preti et al., 2023; Yuan et al., 2021) pay more attention to partition principle, but ignore subgraph-level symmetry analysis. This limitation results in failing to achieve the desired powerful representation ability, namely subgraph degradation. We can easily observe the examples in biochemistry domain, e.g., position isomerism is a common phenomenon in chemistry, which can be viewed as molecular descriptions with the same functional groups compositions but different positions. It is shown in the bottom panel of Figure 1.

We rethink this observation via permutation equivariance and invariance analysis. We can denote the invariant learning ignoring subgraph-level symmetry as $f : f(\tau \cdot G_S) = f(G_S)$, where $\tau$ denotes the exact permutation of subgraph. In other words, the order of subgraphs is independent of the representation of original graph. Obviously, this design is not appropriate for real-world tasks of subgraph learning. Equivariant learning $g : g(\tau \cdot G_S) = \tau \cdot g(G_S)$ at the subgraph-level can have more intuitive interpretability (Bevilacqua et al., 2022).

### 3.3 UNIFYING SUBGRAPH LEARNING VIA RECONSTRUCTION ABILITY

Based on the fresh review of prior practices, we summarize the root reason for subgraph degradation to two limitations as below.

**Node-based subgraphs with redundant information.** The most notable characteristic of the subgraphs $G_S$ extracted by the node-based methods is that there are numerous pairwise subgraphs with overlapping information, namely $G_S^i \cap G_S^j \neq \emptyset$ for $i \neq j$. We also note that the degradation of message passing space (neighbors information) is the main cause of node-based subgraphs failure. However, overlapping subgraphs precisely give each node an independent and 1-WL-similar messaging passing space. We also can find successful practice to support this attitude. ESAN (Bevilacqua et al., 2022) extract subgraph set $G_S^m \subseteq G_S^n$ via a stochastic sampling strategy, which has been widely followed (Zhang et al., 2023). Actually, this design can be viewed as a manner to avoid degradation by reducing overlap between subgraphs. Thus, the redundant information of $G_S$ becomes a key challenge for node-based strategies.

**Partition-based subgraphs with absent information.** We can also summarize the characteristics of $G_S$ obtained by partition-based subgraph selection policy, i.e., (i) there is no information overlap between any pair of subgraphs, namely $G_S^i \cap G_S^j = \emptyset$ for $i \neq j$, (ii) $G_S$ is disordered with no positional associations among subgraphs. The above two indicate that the elements in $G_S$ are independent of each other. Thus, it is obvious that the absence of subgraph-level order information inevitably leads to the failure of distinguishing.

To summarize, we attribute the limitations of above two strategies to redundancy and absence, respectively. The redundant and insufficient subgraphs results in failing to perfectly reconstruct the whole graph. Inspired by this discovery, we take the perspective of perfect reconstruction to remedy above two limitations.

**Definition 1.** *Let subgraph set $G_S$ be extracted by $G$. The subgraph set $G_S$ will be equipped with **reconstruction ability** if there exist a reconstruction function $r(\cdot)$ satisfying $r(G_S) = G$. Specifically, $G_S$ with **redundant reconstruction ability** is defined*

$$r(G_{S^*}) = G \quad \exists G_{S^*} \subset G_S, r(\cdot) \tag{5}$$

*$G_S$ with **perfect reconstruction ability** is defined*

$$r(G_{S^*}) \neq G \quad \forall G_{S^*} \subset G_S, r(\cdot) \tag{6}$$

*If there does not existence a reconstruction function $r(\cdot)$ satisfying $r(G_S) = G$, $G_S$ is with **no reconstruction ability**.*

In view of the limitations of these two methods in reconstruction ability, we expect to extract subgraphs with perfect reconstruction ability to potentially avoid the degradation trap. Since the subgraphs extracted by node-based strategy have irregular information overlap and limited interpretability, we thus pay our attention to improving partition-based subgraph learning.

## 4 RayE-Sub: Rayleigh-resistance Equivariant Subgraph Learning

In this section, we design a powerful architecture *RayE-Sub* to extract subgraphs with perfect reconstruction ability. And, the expressive power of *RayE-WL* is also studied.

### 4.1 Rayleigh-resistance Policy for Subgraph Partition

Subgraph partition methods aim to find the significant boundary, which composes of a series of edges connecting two irrelevant nodes. In implementations, topological characteristics and feature contents are vital factors affecting the effectiveness of partition principles. Inspired by the great superiority of spectral theory in drawing graphs (Spielman, 2019; Kreuzer et al., 2021), we exploit the idea of spectrum to realize the subgraph partitions with topological and feature information.

The Laplacian operator $L_G$ is the entry to spectral theory, where *Rayleigh quotient* of $L_G$ elegantly depict the stability of graph $G$ (Spielman, 2019).

**Definition 2.** *The Rayleigh quotient $q(G)$ of the Laplacian matrix $L_G$ is defined as*

$$q(G) = x^T L_G x = \sum_{(u,v) \in E} w_{uv} (x(u) - x(v))^2, \; x^T x = 1 \tag{7}$$

*where $x \in \mathbb{R}^{n \times 1}$ is the feature matrix of nodes* [1].

Empirically, the value of $q(G)$ is the resonance of both structure $w_{uv}$ and feature contents $(x(u) - x(v))^2$ for all edges $(u, v)$. In other words, *Rayleigh quotient* quantifies the stability of the graph with structure and feature information. The smaller $q(G)$ refers to that nodes are closer with each other so that the graph is more stable. Otherwise, the graph is fragile.

There exists an equation between the Laplacian matrix $L_{G_S^i} \in \mathbb{R}^{n \times n}$ of the subgraph $G_S^i$, boundary matrix $B_{G_S^i} \in \mathbb{R}^{n \times n}$ and extraction matrix $L_G(V_i, V_i) \in \mathbb{R}^{n \times n}$ of $L_G$: $L_G(V_i, V_i) = L_{G_S^i} + B_{G_S^i}$, where $B_{G_S^i}$ represents the boundary between $G_S^i$ and the rest of the graph with $B_{G_S^i}(a, a) = \sum_{b \notin G_S^i} w_{ab}$. For the boundary of global graph $B_G = \sum_{i \in [k]} B_{G_S^i}$, $x^T L_G x = \sum_{i \in [k]} x^T L_{G_S^i} x + x^T B_G x$. Given a specific $x$, $x^T L_G x$ becomes fixed. Thus, discovering partition boundary $\max_{B_G} x^T B_G x$ is equivalent to optimizing $\min_{G_S} q(G_S) = \min_{G_S} \sum_{i \in [k]} x^T L_{G_S^i} x$.

However, this optimization tends to obtain the boundary $B_G$ consisting of all the edges, resulting in a bag of single-node subgraphs. The direct solution is to take the number of nodes as a one of description factors of subgraph stability. Follow this idea, instead of $L_G$, we utilize the normalized Laplacian matrix $N_G = D^{-1/2} L_G D^{-1/2}$ to rewrite the formation of *Rayleigh quotient* as,

$$q^*(G) = x^T N_G x = \sum_{(a,b) \in E} w_{ab} \left( \frac{x(a)}{\sqrt{d_a}} - \frac{x(b)}{\sqrt{d_b}} \right)^2 \tag{8}$$

where $d_j$ represents the degree of $j$. It effectively avoids single-node subgraphs case, since a bag of single-node subgraphs means that all nodes have degrees of 0, which results in infinity $q^*(G)$. Thus, our following discussion is based on the rewritten $q^*(G)$. We provide a theoretical analysis of Equation 8 in Appendix C.5.

Optimizing $\min_{G_S} q^*(G_S)$ is a NP (Non-deterministic Polynomial) problem, thus such optimization is an inaccessible target in practical implementation. To this end, we borrow the resistance distance to implement the quantification defined by *Rayleigh quotient*. Resistance distance (RD) is a basic

---

[1] We let the feature dimension be 1 to simplify subsequent analysis.

metric in graph spectral theory (Xiao & Gutman, 2003). It reflects both distance and the accessibility (number of pathways) between two nodes, which has the potential to characterize the global structural topology. We propose to exploit RD as an alternative to quantify the stability of subgraph.

**Definition 3.** *The resistance distance between two vertices $u$ and $v$ in an electrical network, is measured by the resistance of the entire network when we treat it as one complex resistor. It can be computed by*

$$RD_{uv} = (\delta_u - \delta_v)^T L^+ (\delta_u - \delta_v) \tag{9}$$

*where $\delta_j$ is the elementary unit vector with $\mathbf{1}$ in coordinate $j$ and $L^+$ is the pseudo-inverse of $L$.*

**Theorem 2.** *(Equivalence between Resistance distance and Rayleigh quotient.) Let $u$ and $v$ be any two vertices connected by an edge. Under the accessible constraints, the resistance distance between $u$ and $v$ is approximately equivalent to the stability of pairwise vertices defined by Rayleigh quotient.*

*Proof.* We prove that we can achieve their equivalence by integrating the feature information $x$ into the topology with the edge reweighting strategy of $w_{uv} = ||x_u - x_v||^{-2}$. Detailed proof have been provided in Appendix C.4. Besides, we also provide an efficient approximation approach for calculating RD with the time complexity of $\mathcal{O}(|E|)$ in Appendix E.

## 4.2 SIAMESE-QUERY NETWORK FOR SUBGRAPH-LEVEL EQUIVARIANT AGGREGATION

Above discussion supports extracting a bag of subgraphs without overlap (i.e., 'perfect'). After that, we must guarantee ordered reconstruction of subgraphs. Specifically, we propose a *Siamese-Query* scheme to realize equivariant subgraph learning, as shown in Figure 2. These layers map bags of subgraphs into representation $\mathbf{Z}$ as follows,

$$h_G = L(A, X), \ h_S = \text{CONCAT}[L(A_i, X_i)] \quad Q = h_G W_Q, \ K = h_S W_K, \ V = h_S W_V \tag{10}$$

$$\mathbf{Z} = \text{softmax}(\frac{\text{QK}^\text{T}}{\sqrt{\text{d}}})\text{V} \tag{11}$$

where $W_Q, W_K, W_V \in \mathbb{R}^{d \times d}$ are learnable parameters. $h_G \in \mathbb{R}^{1 \times d}$ and $h_S \in \mathbb{R}^{m \times d}$ respectively denote the graph-level and subgraph-level embedding obtained by an MPNN encoder $L$. The representation of $G$ for prediction denotes $\mathbf{Z}$. *Query* mechanism introduces attention score (order) to each subgraph by mirroring global information, which smoothly realizes subgraph-level equivariant learning. We define $g(h_G, h_S) = \text{softmax}(\frac{\text{QK}^\text{T}}{\sqrt{\text{d}}})$. For any permutation $\tau$ acting on subgraphs, $g(h_G, \tau \cdot h_S) = \tau \cdot g(h_G, h_S)$ is always hold on.

## 4.3 LEARNING OBJECTIVE

*RayE-Sub* is a two-stage learning architecture, where *Rayleigh-resistance* realizes subgraph partition and *Siamese-Query* achieves subgraph-level equivariant learning.

***Rayleigh-resistance*** aims to obtain boundary $B_G$ to partition graph $G$. Specifically, we employ the resistance distance between two connected nodes to quantify the stability of this edge $s(u, v)$. Similar to *Rayleigh quotient*, smaller resistance distance indicates a more stabler connection. Therefore, $B_G$ is composed of the edges $(u, v)$ belonging to top-$\beta$(S), which picks out the top $\beta$ larger of $S$.

$$B_G := \bigcup_{(u,v) \in E} \{(u, v)\}, \ s(u, v) \in \text{top-}\beta(\text{S}) \tag{12}$$

***Siamese-Query*** is the prediction module of subgraph equivariant learning. For each graph with label $Y_i$ and its prediction $\hat{Y}_i$, we impose the cross entropy loss on all $N$ graphs as the learning objective, i.e.,

$$\mathcal{L} := -\frac{1}{N} \sum_{i=1}^{N} Y_i \log(\hat{Y}_i) \tag{13}$$

Table 1: Performance comparisons. The best results are in **bold** and the second best is underlined.

| | MOLHIV | BBBP | SIDER | MUTAG | BA-2Motifs | Spurious-Motif (Sp-M) | | |
| | | | | | | 0.5 | 0.7 | 0.9 |
|---|---|---|---|---|---|---|---|---|
| GCN | $75.5 \pm 1.6$ | $65.3 \pm 1.9$ | $52.1 \pm 2.0$ | $83.7 \pm 4.7$ | $86.8 \pm 1.7$ | $33.2 \pm 1.8$ | $31.6 \pm 1.7$ | $29.6 \pm 6.2$ |
| Graph-SAGE | $74.8 \pm 3.4$ | $64.1 \pm 2.8$ | $52.5 \pm 1.6$ | $84.6 \pm 5.3$ | $85.7 \pm 2.3$ | $34.8 \pm 2.0$ | $31.5 \pm 2.5$ | $30.4 \pm 3.4$ |
| GIN | $75.8 \pm 1.3$ | $66.4 \pm 2.0$ | $56.2 \pm 1.6$ | $89.4 \pm 5.6$ | $89.5 \pm 2.1$ | $39.9 \pm 1.3$ | $39.0 \pm 1.6$ | $38.6 \pm 2.3$ |
| ESAN | $77.2 \pm 1.3$ | $68.8 \pm 1.3$ | $58.1 \pm 1.8$ | $92.0 \pm 5.0$ | $92.9 \pm 2.9$ | **$56.1 \pm 1.7$** | $47.9 \pm 1.5$ | $44.8 \pm 2.9$ |
| GNN-AK | $76.8 \pm 1.2$ | $67.7 \pm 4.2$ | $57.5 \pm 1.4$ | $92.3 \pm 6.8$ | $91.6 \pm 3.3$ | $54.2 \pm 1.2$ | $44.8 \pm 1.7$ | $42.6 \pm 1.8$ |
| SUN | $76.6 \pm 0.9$ | $66.4 \pm 1.5$ | $56.7 \pm 2.0$ | $94.7 \pm 5.2$ | $93.6 \pm 4.1$ | $55.6 \pm 3.2$ | $45.2 \pm 2.4$ | $43.2 \pm 1.6$ |
| IB-subgraph | $76.4 \pm 2.6$ | $68.1 \pm 1.1$ | $57.7 \pm 2.1$ | $91.1 \pm 6.4$ | $90.1 \pm 6.5$ | $54.4 \pm 7.0$ | $48.5 \pm 5.8$ | **$46.2 \pm 5.7$** |
| GSAT | $76.5 \pm 1.5$ | $69.0 \pm 1.2$ | $57.2 \pm 1.3$ | **$96.7 \pm 2.1$** | $97.4 \pm 1.9$ | $46.6 \pm 2.9$ | $49.1 \pm 3.0$ | $39.8 \pm 2.4$ |
| DIR | $76.3 \pm 1.1$ | $68.2 \pm 1.4$ | $57.8 \pm 1.8$ | $92.1 \pm 2.3$ | $93.8 \pm 9.6$ | $45.5 \pm 3.8$ | $41.1 \pm 2.6$ | $37.6 \pm 2.0$ |
| RayE-Sub | **$77.6 \pm 1.0$** | **$72.2 \pm 1.1$** | **$58.4 \pm 1.9$** | $95.6 \pm 2.4$ | **$98.5 \pm 1.0$** | $53.8 \pm 2.0$ | **$49.6 \pm 2.9$** | $45.8 \pm 2.2$ |

### 4.4 THE EXPRESSIVE POWER OF RAYE-WL

We introduce WL analogue (RayE-WL) for RayE-Sub to support our next study of the expressiveness. Due to limited pages, we only present the core step of RayE-WL, color refinement algorithm, detailed algorithm is provided in Algorithm 3. On subgraph $G_S^i \in G_S$, the color of node $v \in G_S^i$ is refined according to the rule,

$$c_{v,G_S^i}^{t+1} := \text{HASH}(c_{v,G_S^i}^t, N_{v,G_S^i}^t, c_{G_S^i}^{t+1}) \tag{14}$$

$$c_{G_S^i}^{t+1} := \text{HASH}(c_{G_S^i}^t, M_{G_S^i,G}^t, c_G^{WL,t}) \tag{15}$$

$N_{v,G_S^i}^t$ denotes the multiset of colors in $v$'s neighborhood over subgraph $G_S^i$ after the $t$-th iteration. $c_{G_S^i}^t$ represents the color of the subgraph $G_S^i$ in which node $v$ is located after the $t$-th iteration. $M_{G_S^i,G}^t$ denotes the color multiset of all subgraphs of the graph $G$ independently mapped by 1-WL after the $t$-th iteration, $M_{G_S^i,G}^t = \{c_S^{\text{WL},t} | S \in G_{G_S^i}\}$. $c_G^t$ represents the color of the graph $G$ mapped by 1-WL after the $t$-th iteration.

**Theorem 3.** *(RayE-WL is more powerful than 1-WL.) RayE-WL is strictly more powerful than 1-WL in distinguishing between non-isomorphic graphs, which is upper bounded by 3-WL.*

*Proof.* Given two non-isomorphic graphs $G$, $H$, we first prove that RayE-WL is stronger than 1-WL by two terms. (i) If they can be distinguished by the 1-WL graph isomorphism test, RayE-WL can strictly distinguish them. (ii) There exist graphs that cannot be distinguished by 1-WL but can be distinguished by RayE-WL. Then, we comprehensively compare our architecture with 3-WL. Detailed proof is provided in Appendix D.2.

## 5 EXPERIMENTS

### 5.1 EXPERIMENTAL SETTINGS

**Datasets.** The datasets for evaluation are two-fold, four real-world datasets and two synthetic datasets on graph classification tasks. **(i) Real-world datasets:** MUTAG (Debnath et al., 1991) and three OGB datasets (Hu et al., 2020) (MOLHIV, BBBP and SIDER). **(ii) Synthetic datasets:** BA-2Motifs (Luo et al., 2020) and Spurious-Motif (Sp-M) (Wu et al., 2022). The detailed dataset statistics are provided in Appendix F.1.

**Baselines.** Our baselines are three-fold, including GNN backbones, node-based subgraph learning methods and partition-based subgraph learning models. **(i) Backbone baselines:** GCN Kipf & Welling (2016), Graph-SAGE Hamilton et al. (2017) and GIN Xu et al. (2018). **(ii) Node-based subgraph learning methods:** ESAN Bevilacqua et al. (2022), GNN-AK Zhao et al. (2021), SUN Frasca et al. (2022). **(iii) Partition-based subgraph learning models:** IB-subgraph Yu et al. (2020), GSAT Miao et al. (2022) and DIR Wu et al. (2022). The details of baselines can be found in Appendix F.2

**Backbone and Metrics.** We exploit GIN as the backbone of *RayE-Sub* due to its extensive popularity. To ensure fair comparisons, we let GIN serve as the basic model in all baselines. We also explore the influence of different backbones on performance, which is provided in Section 5.3. For prediction performance, we employ ROC-AUC for OGB datasets (MOLHIV, BBBP and SIDER) and accuracy for the other datasets.

Table 2: Ablation results. The performance of *RayE-Sub* with different backbones.

|  | RayE-Sub (GIN) | RayE-Sub (PNA) |
|---|---|---|
| Sp-M (0.5) | $53.8 \pm 2.0$ | $\mathbf{72.7 \pm 2.6}$ |
| Sp-M (0.7) | $49.6 \pm 2.9$ | $\mathbf{66.7 \pm 1.9}$ |
| Sp-M (0.9) | $45.8 \pm 2.2$ | $\mathbf{60.6 \pm 2.3}$ |
| MUTAG | $\mathbf{95.6 \pm 2.4}$ | $94.8 \pm 2.8$ |
| SIDER | $\mathbf{58.4 \pm 1.9}$ | $57.3 \pm 2.0$ |

## 5.2 MAIN RESULTS

The overall performance is summarized Table 1, and we have the following **Obs**ervations:

**Obs 1: Consistently outperform traditional backbone models on all datasets.** Compared with traditional backbone methods, our approach achieves significant improvements across all datasets with a maximum performance margin of 9%. This improvement empirically verifies that subgraph learning can effectively boost the expressive power of graph learning.

**Obs 2: Compared with other subgraph learning methods, *RayE-Sub* achieves competitive results in both real and synthetic datasets.** Encouragingly, *RayE-Sub* obtains the SOTA on five datasets. Specifically, our *RayE-Sub* outperforms best baselines by 3.2% and 1.1% respectively on BBBP and BA-2Motifs. Such performance superiority can be explicitly attributed to the coupling effects of both two objectives, i.e., *Rayleigh-resistance* based subgraph partition and *Siamese-Query* based equivariant subgraph learning.

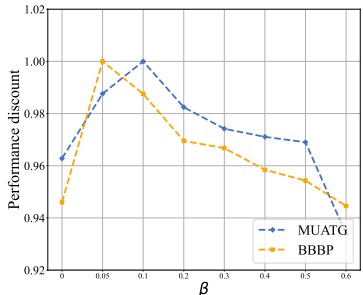

Figure 3: Sensitivity analysis of $\beta$ on MUTAG and BBBP, where $y$-$axis$ represents the discount of each setting to the best performance.

**Obs 3: Partition-based subgraph discovery approaches have better predictive ability than node-based methods.** In all eight tests, partition-based subgraph learning methods including *RayE-Sub* obtain seven best results. Especially, the top-2 performances on all datasets are almost achieved by partition-based approaches. These results verify that partition-based methods not only have better interpretability, but also are with better expressiveness.

## 5.3 ABLATION STUDY AND SENSITIVITY ANALYSIS

Our ablation studies are designed to explore the effects of *RayE-Sub* with different backbones. Specifically, we compare the performances of *RayE-Sub* when GIN and PNA are utilized as base encoder respectively. As shown in Table 2, we observe that PNA-based architecture has strong expression ability on synthetic datasets. We analyze that the design of PNA using multiple aggregators can better exploit common sub-units of graphs, which matches the generation principles of synthetic datasets (Wu et al., 2022). GIN has an advantage on more complex tasks in real-world datasets.

Furthermore, we conduct sensitivity analysis of model performance about $\beta$, the results are in Figure 3. Our model achieves best results when $\beta$ is within $[0.05, 0.1]$. In other words, about 5%-10% of the edges in two molecular datasets are with fragile connections, and their role is simply to bond different functional groups. Therefore, we set $\beta = 0.05$ on all datasets. We also provide visualizations of the subgraphs discovered by *Rayleigh-resistance* to verify such observation in Appendix G.

## 6 CONCLUSION

In this paper, we systematically study subgraph learning methods from the perspective of perfect subgraph reconstruction, and propose a novel architecture *RayE-Sub*. We first observe and summarize the phenomenon of subgraph degradation of current subgraph learning methods. We then exploit the spectral theory and subgraph equivalence principle to respectively remedy overlapping and disorder issues, which jointly contribute to perfect reconstruction. Experiments on both synthetic and real-world datasets demonstrate the effectiveness of *RayE-Sub*. Moreover, theoretical analysis and practical observations profoundly guarantee the superiority of our architecture.

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

## A   MORE OBSERVATIONS OF SUBGRAPHS FAILURE

**Example 1.** *(**The degeneration of Node-based subgraphs.**) Let $G_1 = \{V_1, E_1\}$ and $G_2 = \{V_2, E_2\}$ be a pair of graphs with $n = 2kl + 1$, where $k, l$ are two positive integers satisfying $kl > 3$. Note that $V_1 = V_2 = [n]$, $E_1$ and $E_2$ satisfy the following conditions,*

$$E_1 = \{\{i, (i \bmod kl) + 1\} : i \in [kl]\} \cup \{\{i + kl, (i \bmod kl) + kl + 1\} : i \in [kl]\} \\ \cup \{\{n, i\} : i \in [2kl], i \bmod l = 0\} \tag{16}$$

$$E_2 = \{\{i, (i \bmod 2kl) + 1\} : i \in [2kl]\} \cup \{\{n, i\} : i \in [2kl], i \bmod l = 0\} \tag{17}$$

By setting $m = 2, k = 2$, we visualize a specific example in the top panel of Figure 1. Since EGO is more accessible for readers than other node-based policies. Next, we provide more observations of subgraphs failure on ND-based (Node Delete) and NM-based (Node Marking) strategies. Actually, our Example 1 also can support the failure of ND-based and NM-based subgraph learning. We confirm this fact via an intuitive case. Specifically, we can obtain $G$ and $H$ by setting $m = 1, k = 4$ as shown in Figure 4. For original $G$ and $H$, they cannot be distinguished by 1-WL (Zhang et al., 2023). Therefore, we focus more on explore whether we can distinguish them by node-based subgraphse. We can extract their subgraph set from $G$ and $H$ respectively based on ND policy as shown in Figure 4. Unfortunately, they still can't be distinguished by subraph-level analysis. We will explain the reasons in detail in the following analysis.

The $G_S^1$ extracted from $G$ is two 4-cycle regular graphs, while the $H_S^1$ extracted from $H$ is a 8-cycle regular graph. It is obvious that 3-WL could not distinguish them. Although 3-WL can distinguish between them well, existing methods do not have this ability. On the one hand, almost all node-based subgraph learning methods are upper bounded by 3-WL. They are unable to distinguish $G_S^1$ and $H_S^1$. For $G_S^2$ and $H_S^2$, similar to $G_S^1$, still can be distinguished by 3-WL, but not by 1-WL. Intuitively, the neighbor information of each node in $G_S^2$ and $H_S^2$ is same. This homogenization will result in their indistinguishability. Theoretically, we can confirm this fact with the simple aggregation practice of Algorithm 1. The failure of NM-based method is similar to ND-based strategy, so we don't repeat it.

## B   MORE ANALYSIS OF SPECTRAL THEORY

It is a traditional research direction to draw the distance between nodes, compare graphs and partition substructures from the view of spectral theory. In this section, we provide more analysis of spectral theory to better support our research motivation and specific methods.

### B.1   UNDERSTANDING RAYLEIGH QUOTIENT FROM SPECTRAL THEORY.

Fundamentally, our subgraphs partition method based on Rayleigh quotient is inspired by spectral graph theory. As understood in the text, Rayleigh quotient intuitively show the stability of graphs. More importantly, Rayleigh quotient has a rich theoretical basis to guide subgraph partition.

**Lemma 2.** *(Spielman, 2019) Let $M$ be a symmetric matrix with eigenvalues $\mu_1, ..., \mu_n$ and a corresponding orthonormal basis of eigenvectors $\varphi_1, ..., \varphi_n$. Let $x$ be a vector whose expansion in the eigenbasis is*

$$x = \sum_i c_i \varphi_i \tag{18}$$

*Then,*

$$x^T M x = (\sum_i c_i \varphi_i)^T M (\sum_j c_j \varphi_j) = (\sum_i c_i \varphi_i)^T (\sum_j c_j \mu_j \varphi_j) = \sum_{i,j} c_i c_j \varphi_i^T \varphi_j = \sum_i c_i^2 \mu_i \tag{19}$$

*as*

$$\varphi_i^T \varphi_i = \begin{cases} 1, & i = j \\ 0, & i \neq j \end{cases} \tag{20}$$

**Lemma 3.** *(Spielman, 2019) Let $M$ be an n-dimensional real symmetric matrix. There exist numbers $\mu_1, ..., \mu_n$ and orthonormal vectors $\varphi_1, ..., \varphi_n$ such that $M\varphi_i = \mu_i \varphi_i$. Moreover,*

$$\varphi_1 \in \arg \max_{||x||=1} x^T M x \tag{21}$$

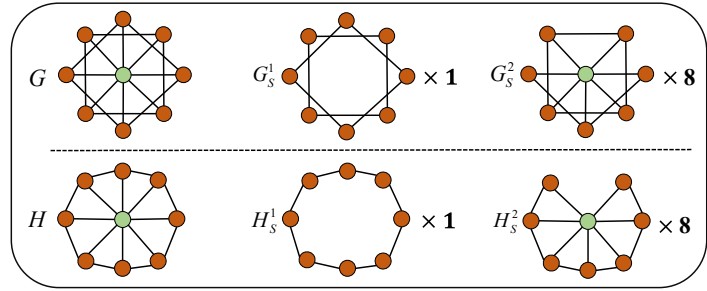

Figure 4: The support graphs ($G$ and $H$) of ND-based subgraph learning failure by setting $m = 1, k = 4$ in Example 1.

*and for $2 \leq i \leq n$*

$$\varphi_i \in \arg \max_{\substack{||x||=1 \\ x^T \varphi_j=0, j<i}} x^T M x \tag{22}$$

*Similarly,*

$$\varphi_i \in \arg \min_{\substack{||x||=1 \\ x^T \varphi_j=0, j>i}} x^T M x \tag{23}$$

**Theorem 4.** *(Spielman, 2019) Let M be a symmetric matrix with eigenvalues $\mu_1 \geq \mu_2 \geq \cdots \geq \mu_n$. Then,*

$$\mu_k = \max_{\substack{S \subseteq \mathbb{R}^n \\ \dim(S)=k}} \min_{\substack{x \in S \\ x \neq 0}} \frac{x^T M x}{x^T x} = \min_{\substack{T \subseteq \mathbb{R}^n \\ \dim(T)=n-k+1}} \max_{\substack{x \in T \\ x \neq 0}} \frac{x^T M x}{x^T x} \tag{24}$$

*where the maximization and minimization are over subspaces $S$ and $T$ of $\mathbb{R}^n$.*

When $M$ is the Laplacian operator $L_G$ of the graph $G$, this theorem reveals a fact that the extreme values of *Rayleigh quotient* are equivalent with eigenvalues of $L_G$. We provide two lines of proof to support this theorem.

***Proof 1.*** Let $\varphi_1, ..., \varphi_n$ be an orthonormal basis of eigenvectors of $M$ corresponding to $\mu_1, ..., \mu_n$. We will just verify the first characterization of $\mu_k$. First, let $S$ be the span of $\varphi_1, ..., \varphi_k$. We can expand every $x \in S$ as

$$x = \sum_{i=1}^{k} c_i \varphi_i \tag{25}$$

Based on Lemma 2, we can obatin

$$\frac{x^T M x}{x^T x} = \frac{\sum_i c_i^2 \mu_i}{\sum_i c_i^2} \geq \frac{\sum_i c_i^2 \mu_k}{\sum_i c_i^2} = \mu_k \tag{26}$$

Thus,

$$\min_{x \in S} \frac{x^T M x}{x^T x} \geq \mu_k \tag{27}$$

To show that this is in fact the maximum, we will prove that for all subspaces $S$ of dimension $k$,

$$\min_{x \in S} \frac{x^T M x}{x^T x} \leq \mu_k \tag{28}$$

Let $T$ be the span of $\varphi_1, ..., \varphi_n$. As $T$ has dimension $n - k + 1$, every $S$ of dimension $k$ has an intersection with $T$ of dimension at least 1. Thus,

$$\min_{x \in S} \frac{x^T M x}{x^T x} \leq \min_{x \in S \cap T} \frac{x^T M x}{x^T x} \leq \max_{x \in T} \frac{x^T M x}{x^T x} \tag{29}$$

Any $x$ in $T$ can be expressed as

$$x = \sum_{i=k}^{n} c_i \varphi_i \tag{30}$$

and so for $x$ in $T$

$$\frac{x^T M x}{x^T x} = \frac{\sum_{i=k}^{n} c_i^2 \mu_i}{\sum_{i=k}^{n} c_i^2} \leq \frac{\sum_{i=k}^{n} c_i^2 \mu_k}{\sum_{i=k}^{n} c_i^2} = \mu_k \tag{31}$$

In conclusion, we prove $\mu_k = \max_{\substack{S \subseteq \mathbb{R}^n \\ \dim(S)=k}} \min_{\substack{x \in S \\ x \neq 0}} \frac{x^T M x}{x^T x}$, the other is similar.

***Proof 2.*** First, we recall that the gradient of a function at its maximum must be the zero vector. Let's compute that gradient. We have

$$\nabla x^T x = 2x \tag{32}$$

and

$$\nabla x^T M x = 2M x \tag{33}$$

Thus,

$$\nabla \frac{x^T M x}{x^T x} = \frac{(x^T x)(2M x) - (x^T M x)(2x)}{\left(x^T x\right)^2} \tag{34}$$

In order for this to be zero, we must have

$$(x^T x)(2M x) = (x^T M x)(2x) \tag{35}$$

which is equivalent to

$$M x = \frac{x^T M x}{x^T x} x \tag{36}$$

That is, if and only if $x$ is an eigenvector of $M$ with eigenvalue equal to its Rayleigh quotient. As $x$ maximizes the Rayleigh quotient, this eigenvalue must be $\mu_1$. Based on Lemma 3, it is not hard to prove that Theorem 4 by generalizing this characterization to all of the eigenvalues of $M$. Much of the above discussion is derived from (Spielman, 2019), and more detailed understanding is also available from it.

### B.2 FURTHER ANALYSIS OF RESISTANCE DISTANCE.

The resistance distance between two vertices $a$ and $b$ in an electrical network is the resistance of the entire network when we treat it as one complex resistor. That is, we consider an electrical flow that sends one unit of current into node $a$ and outflows one unit of current from node $b$. We then measure the potential difference between $a$ and $b$ that is required to realize this current, define it to be the resistance distance between $a$ and $b$, and write it $RD_{ab}$.

$$RD_{ab} \overset{def}{=} (\delta_a - \delta_b)^T L^+ (\delta_a - \delta_b) \tag{37}$$

where $\delta_j$ is the elementary unit vector with $\mathbf{1}$ in coordinate $j$ and $L^+$ is the pseudo-inverse of $L$.

$$\begin{aligned}
RD_{ab} &\overset{def}{=} (\delta_a - \delta_b)^T L^+ (\delta_a - \delta_b) \\
&= (L^{+/2}(\delta_a - \delta_b)) L^{+/2}(\delta_a - \delta_b) \\
&= ||L^{+/2}(\delta_a - \delta_b)||^2 \\
&= ||L^{+/2}\delta_a - L^{+/2}\delta_b||^2 \\
&= \text{dist}(L^{+/2}\delta_a, L^{+/2}\delta_b)^2
\end{aligned} \tag{38}$$

**Resistance Distance through Energy Minimization.** Physics tells us that the vertices will settle into the position that minimizes the potential energy. The potential energy of an ideal linear spring with constant $w$ when stretched to length $l$ is

$$\frac{1}{2}wl^2 \tag{39}$$

So, the potential energy in a configuration $x$ is given by

$$\xi(x) \overset{def}{=} \frac{1}{2} \sum_{(a,b)\in E} w_{ab}(x(a) - x(b))^2 \tag{40}$$

The lowest energy must be reached when the each variable of $\xi(x)$ is zero. The partial derivative with respect to $x(a)$ is

$$\frac{1}{2} \sum_{(a,b)\in E} 2w_{ab}(x(a) - x(b)) = \sum_{(a,b)\in E} w_{ab}(x(a) - x(b)) \tag{41}$$

Setting this to zero gives the equations we can obtain

$$x(a) = \frac{1}{d_a} \sum_{(a,b)\in E} w_{ab}x(b) \tag{42}$$

For future reference, we state this result as a lemma (Spielman, 2019).

**Lemma 4.** *(Spielman, 2019) Let $G = (V, E, w)$ be a connected, weighted graph, let $B \subset V$, and let $S = V - B$. Given $x(B)$, $\xi(x)$ is minimized by setting $x(S)$ so that $x$ is harmonic on $S$.*

We focus on the resistance distance between $a$ and $b$, thus $x$ should be harmonic on $V - \{a, b\}$. Fortunately, we already know how compute such a vector $x$. Set

$$y = L^+(\delta_a - \delta_b)/RD_{ab} \tag{43}$$

We can get

$$y(a) - y(b) = (\delta_a - \delta_b)^T L^+(\delta_a - \delta_b)/RD_{ab} = 1 \tag{44}$$

thus $y$ is harmonic on $V - \{a, b\}$. We further set $x = y - 1y(s)$. It is obvious that $x$ satisfies $x(s) = 0, x(t) = 1$, and it is harmonic on $V - \{a, b\}$. We compute the energy to be

$$\begin{aligned}
x^T L x &= y^T L y \\
&= \frac{1}{(RD(a,b))^2}(L^+(\delta_a - \delta_b))^T L(L^+(\delta_a - \delta_b)) \\
&= \frac{1}{(RD(a,b))^2}(\delta_a - \delta_b)^T L^+ L L^+(\delta_a - \delta_b) \\
&= \frac{1}{(RD(a,b))^2}(\delta_a - \delta_b)^T L^+(\delta_a - \delta_b) \\
&= \frac{1}{RD(a,b)}
\end{aligned} \tag{45}$$

This derivation reveals a fact that the weights of edges are the reciprocals of their resistance distance. In practical implementation, we employ this understanding to achieve the approximate resistance distance with the time complexity of $\mathcal{O}(|E|)$. More importantly, Equation 45 builds a theoretical bridge between *Rayleigh quotient* and *Resistance distance*, which inspires us to utilize resistance distance to describe the stability of the graph and propose *Rayleigh-resistance*.

## C PROOF OF THEOREMS

### C.1 PROOF OF LEMMA

**Lemma 1.** *(Function composition (Arora et al., 2016).)* MPNNs can repeatedly update each node's embedding by aggregating information from their neighbors. The graph-level embedding $h_G$ can be obtained by

$$h_G = \text{POOL} \circ \sigma \circ \text{T}_\text{L} \circ \cdots \circ \text{T}_2 \circ \sigma \circ \text{T}_1(\text{A}, \text{X}) \tag{46}$$

where $\sigma$ and $T_l$ are the $l$-th layer update and aggregation operators (matrices), POOL is the READOUT operation, $\circ$ denotes layer composition. There exists a global aggregation matrix T, merge from $\{T_i | i = 1, 2, ..., l\}$, to make this architecture have equivalent representation:

$$h_G = \text{POOL} \circ \sigma \circ \text{T}(A, X) \tag{47}$$

where each element $\text{T}(i, j)$ indicates the aggregation coefficient from $j$ to $i$.

*Proof.* It is clear that any function represented by a ReLU DNN is a continuous piecewise linear (PWL) function. To see the converse, we first note that any PWL function can be represented as a linear combination of piecewise linear convex functions (Wang & Sun, 2005). For every piecewise linear function $h_G : \mathbb{R}^n \to \mathbb{R}$, there exists a finite set of affine linear functions $h_1, ..., h_k$ and subsets $S_1, ..., S_p \subseteq \{1, ..., k\}$ (not necessarily disjoint) where each $S_i$ is of cardinality at most $n + 1$, such that

$$h_G = \sum_{j=1}^{p} s_j (\max_{i \in S_j} h_i) \tag{48}$$

where $s_j \in \{-1, +1\}$ for all $j = 1, ..., p$. Since a function of the form $\max_{i \in S_j} h_i$ is a piecewise linear convex function with at most $n + 1$ pieces (because $|S_j| \leq n + 1$), Equation 48 says that any continuous piecewise linear function (not necessarily convex) can be obtained as a linear combination of piecewise linear convex functions each of which has at most $n + 1$ affine pieces. (Arora et al., 2016) has shown that composition, addition, and pointwise maximum of PWL functions are also representable by ReLU DNNs. In particular, we note that $\max\{x, y\} = \frac{x+y}{2} + \frac{|x-y|}{2}$ is implementable by a two layer ReLU network and use this construction in an inductive manner to show that maximum of $n + 1$ numbers can be computed using a ReLU DNN with depth at most $\lceil \log_2(n + 1) \rceil$. We can conclude that every piecewise linear function $h_G$ can be represented by a function with at most $\lceil \log_2(n + 1) \rceil$ depth. Thus, there exists

$$\sigma \circ \text{T}_L \circ \cdots \circ \text{T}_2 \circ \sigma \circ \text{T}_1 = \text{T} \tag{49}$$

We finish this proof.

## C.2 PROOF OF THEOREMS 1

**Theorems 1.** *(The degradation of nsode-based subgraph learning.)* *Suppose the ground-truth aggregation matrix of graph $G$ is $M$, the aggregation result from node-level MPNNs is $T_G$, and the result from node-based subgraph learning is $T_S$. Then, there exists graph $G$ satisfying $T_S = T_G$.*

*Proof.* Node-based subgraph learning generate $n$ subgraphs $G_S = \{G_S^1, G_S^2, \cdots, G_S^n\}$, wherein every subgraph is associated with a unique node. The aggregation process of each subgraph $G_S^i$ has a corresponding attention $T_S^i$. For all the subgraphs, the attention of information passing between all nodes is $T_S = \sum_{i \in [n]} T_S^i$. Thus, we turn the question into a proof of whether $\sum_{i \in [n]} T_S^i = T_G$ exists.

Since (Frasca et al., 2022) has shown that the ND-based and NM-based methods are the most expressive node-based subgraph learning strategies, we just study ND-based subgraph discovery policy in our discussion. Consider a complete graph $G$ with 5 nodes and its subgraph set $G_S$ as shown in Figure 5, and briefly assume that all node attributes are the same. We observe that each subgraph is a complete graph with 4 nodes, which is not fundamentally different from $G$. And, we can easily deduce that $T_G$ is a constant multiple of $\sum_{i \in [n]} T_S^i$, which doesn't change the final result. Therefore, there exists graph $G$ satisfying $T_S = T_G$.

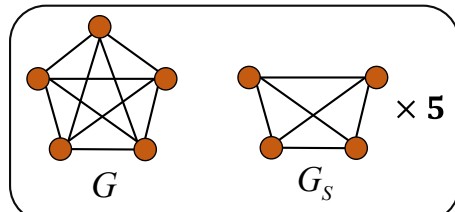

Figure 5: A complete graph $G$ with 5 nodes to support Theorems 1.

Theorem 1 indicates the existence of subgraph degradation, which verifies our intuitive observation. Note that the degradation of subgraphs is not universal, and it doesn't alter the fact that the expression ability of subgraph learning is strictly more powerful than 1-dimensional Weisfeiler-Lehman (1-WL).

### C.3 The Expressive Power of Node-based Methods (Node-WL)

Theorem 1 indicates the existence of subgraph degradation, which verifies our intuitive observation. Note that the degradation of subgraphs is not universal, and it doesn't alter the fact that the expression ability of subgraph learning is strictly more powerful than 1-dimensional Weisfeiler-Lehman (1-WL).

We always analyze that Node-WL is stronger than 1-WL by proving following two terms. Given two non-isomorphic graphs $G$, $H$, (i) if they can be distinguished by the 1-WL graph isomorphism test, Node-WL can strictly distinguish them; (ii) there exist graphs that cannot be distinguished by 1-WL but can be distinguished by Node-WL. We can conclude that there will be some non-isomorphic pairs of graphs, they can neither be distinguished by Node-WL nor 1-WL. The cases of subgraph degradation corresponds to this scenario. Therefore, the cases of subgraph degradation do not affect the proof of these two terms and the expression ability of subgraph learning is still strictly more powerful than 1-WL.

This work aims to systematically study this degradation phenomenon and further provide a more stable subgraph learning architecture to address this subgraph degradation. In terms of expression ability, our architecture is still upper bounded by 3-WL.

### C.4 Proof of Theorems 2

**Theorems 3.** *(**Equivalence between Resistance distance and Rayleigh quotient.**) Let $u$ and $v$ be any two vertices connected by an edge. Under the accessible constraints, the resistance distance between $u$ and $v$ is approximately equivalent to the stability of pairwise vertices defined by Rayleigh quotient.*

*Proof.* Recall that we have $L_G = U^T W U$, where $U \in \mathbb{R}^{m \times n}$ is the signed edge-vertex adjacency matrix and $W \in \mathbb{R}^{m \times m}$ is the diagonal matrix of edge weights. Thus, $N_G$ can be rewritten as $D^{-1/2} U^T W U D^{-1/2}$. Consider the following two derivations,

$$x^T N_G x = x^T D^{-1/2} U^T W U D^{-1/2} x = ||W^{1/2} U D^{-1/2} x||^2 \tag{50}$$

$$RD_{uv} = (\delta_u - \delta_v)^T L_G^+ L_G L_G^+ (\delta_u - \delta_v) = ||W^{1/2} U L_G^+ (\delta_u - \delta_v)||^2 \tag{51}$$

The comparability between Equation 50 and 51 transforms original question into exploring whether there is a method to make $D^{-1/2} x$ and $L_G^+ (\delta_u - \delta_v)$ keep related connection. In other words, we will explore whether there exist the function $\phi(\cdot)$ satisfying $D^{-1/2} x = \phi(L_G^+ (\delta_u - \delta_v))$. We investigate this problem by integrating the feature information $x$ into the topology, and prove that we can find appropriate ways to achieve their equivalence.

Since $L_G$ is symmetric we can diagonalize it and write

$$L_G = \sum_{i=1}^{n-1} \lambda_i \mu_i \mu_i^T \tag{52}$$

where $\lambda_1, \lambda_2, ..., \lambda_{n-1}$ are the nonzero eigenvalues of $L_G$ and $\mu_1, \mu_2, ...., \mu_{n-1}$ are a corresponding set of orthonormal eigenvectors. Thus, we can obtain the $L_G^+$,

$$L_G^+ = \sum_{i=1}^{n-1} \frac{1}{\lambda_i} \mu_i \mu_i^T \tag{53}$$

For all nodes $u$ and $v$ connected by edges, we can easily obtain $(\delta_u - \delta_v) \in \mathbb{R}^{n \times |E|}$. We can derive that the i-th row element of $L_G^+ (\delta_u - \delta_v)$ is denoted as $\sum_{k=1}^{n-1} \frac{1}{\lambda_k} (\mu_{ki}^2 - \mu_{ki} \mu_{kj}) \kappa_i$, where $\kappa_i \in \mathbb{R}^{1 \times n}$, and if $j$ directly connected to $i$, then $\kappa_i[j]$ has a value of 1, otherwise it is 0. For $D^{-1/2} x$, its i-th row element is $\frac{x_i}{\sqrt{d_i}}$. Therefore, we transform this question to explore whether there exist the function $\phi(\cdot)$

satisfying $\frac{x_i}{\sqrt{d_i}} = \phi(\sum_{k=1}^{n-1} \frac{1}{\lambda_k}(\mu_{ki}^2 - \mu_{ki}\mu_{kj})\kappa_i)$. We define $\mathcal{N}(i) = \{j^1, ..., j^{d_i}\}$ and further simplify the latter term:

$$
\begin{aligned}
\sum \sum_{k=1}^{n-1} \frac{1}{\lambda_k}(\mu_{ki}^2 - \mu_{ki}\mu_{kj})\kappa_i &= d_i \sum_{k=1}^{n-1} \frac{1}{\lambda_k}\mu_{ki}^2 - \sum_{j \in \mathcal{N}(i)} \sum_{k=1}^{n-1} \frac{1}{\lambda_k}\mu_{ki}\mu_{kj} \\
&= \left(\sum_{k=1}^{n-1} \frac{1}{\lambda_k}\mu_{ki}^2 - \sum_{k=1}^{n-1} \frac{1}{\lambda_k}\mu_{ki}\mu_{kj^1}\right) + ... + \left(\sum_{k=1}^{n-1} \frac{1}{\lambda_k}\mu_{ki}^2 - \sum_{k=1}^{n-1} \frac{1}{\lambda_k}\mu_{ki}\mu_{kj^{d_i}}\right) \\
&= \sum_{k=1}^{n-1} \frac{1}{\lambda_k}(\mu_{ki}^2 - \mu_{ki}\mu_{kj^1}) + ... + \sum_{k=1}^{n-1} \frac{1}{\lambda_k}(\mu_{ki}^2 - \mu_{ki}\mu_{kj^{d_i}})
\end{aligned}
\tag{54}
$$

We recall the definition of *access time* in graph spectral theory. $H(i, j)$ is the expected number of steps before node $j$ is visited, starting from node $i$.

$$
H(i, j) = 2|E| \sum_{k=1}^{n-1} \frac{1}{\lambda_k}\left(\frac{\mu_{ki}^2}{d_i} - \frac{\mu_{ki}\mu_{kj}}{\sqrt{d_i d_j}}\right)
\tag{55}
$$

The similarity between Equation 54 and 55 inspires us to employ the concept of *access time* to denote $\phi(\cdot)$. Specifically, we define that $\phi(L_G^+(\delta_u - \delta_v)) = \frac{\sum_{j \in N(i)} H(i,j)}{\sqrt{d_i}}$. It is not hard to implement. Actually, the *access time* between neighbors is determined by the weight of the edges and the local topology. Therefore, integrating feature information $x$ into the topology is the key to ensure this equation. The problem has been transformed to prove the correlation between $x_i$ and $\sum_{j \in N(i)} H(i, j)$.

To simplify above discussion, we set the dimension of $x$ to 1. In practical implementation, however, the dimension of $x$ is $d$ (non-zero). Thus, we explore the topological relationship between nodes from the embedded Euclidean space. We first achieve the reweighting of edge via $w_{uv} = ||x_u - x_v||^{-2}$. This design reveals that the isolation or centrality of node embedding in Euclidean space must lead to its topological isolation or centrality. For example, isolated node $i$ ($x_i$) must be hard to reach, so the *access time* ($\sum_{j \in N(i)} H(i, j)$) to its neighbors must also be long, which satisfies our expectation.

In conclusion, when the reweighting strategy of edge $w_{uv} = ||x_u - x_v||^{-2}$ is achieved, the resistance distance between $u$ and $v$ is approximately equivalent to the stability of pairwise vertices defined by Rayleigh quotient. The theoretical understanding of effective resistance distance supports this design: the edge weight and resistance distance have opposite tendency (Spielman, 2019).

## C.5 PROOF OF EQUATION 8

We can rewrite the formation of *Rayleigh quotient*:

$$
q^*(G) = x^T N_G x = \sum_{(a,b) \in E} w_{ab}\left(\frac{x(a)}{\sqrt{d_a}} - \frac{x(b)}{\sqrt{d_b}}\right)^2
\tag{56}
$$

where $d_j$ represents the degree of $j$. Intuitively, Equation 56 effectively avoids single-node subgraphs case by considering the degree of node in subgraph. However, we still wonder if this rewriting theoretically results in different semantics. We make the following derivation:

since

$$
x^T N_G x = x^T D^{-1/2} L_G D^{-1/2} x
\tag{57}
$$

we can obtain

$$
N_G(V_i, V_i) = D^{-1/2} L_{G_S^i} D^{-1/2} + D^{-1/2} B_{G_S^i} D^{-1/2}
\tag{58}
$$

further, we get

$$
x^T N_G x = \sum_{i \in [k]} x^T D^{-1/2} L_{G_S^i} D^{-1/2} x + x^T D^{-1/2} B_G D^{-1/2} x
\tag{59}
$$

Given $x$, $x^T N_G x$ is also fixed. Thus, discovering the significant boundary $\max_{B_G} x^T D^{-1/2} B_G D^{-1/2} x$ is equivalent to optimizing $\min_{G_S} x^T N_G x = \min_{G_S} \sum_{i \in [k]} x^T D^{-1/2} L_{G_S^i} D^{-1/2} x =$

$\min_{G_S} \sum_{i \in [k]} (D^{-1/2}x)^T L_{G_S^i} D^{-1/2}x$. In this case, we can view $y = D^{-1/2}x$ as the feature of nodes. Different from the previous one, topological information (degree) of nodes is also an important standard for graph partition. Therefore, our improvement can not only realize the subgraph partition guided by Rayleigh quotient but also avoid the defect of single-node subgraph case.

## D    THE STUDY OF EXPRESSIVE POWER

### D.1    THE WEISFEILER-LEHMAN ALGORITHM AND ITS VARIANTS

In this section, we first present two families of algorithms $k$-dimensional Weisfeiler-Lehman algorithm ($k$-WL) and $k$-dimensional Folklore WeisfeilerLehman algorithms ($k$-FWL), both parameterized by $k \in [n]$. $k$-WL and $k$-FWL both construct a coloring of $k$-tuples of vertices, where the $k$-tuple $i$ is denoted by $\boldsymbol{i} \in [n]^k$ ( $\boldsymbol{v_i} = (v_{i_1}, ..., v_{i_k}) \in V^k$). And, the output of $k$-WL and $k$-FWL obtain denotes the color $\mathbf{C}_{\boldsymbol{i}}$ of each $k$-tuple $\boldsymbol{v_i}$. The difference of $k$-WL and $k$-FWL is in how the colors are aggregated from neighboring $k$-tuples. We define two notions of neighborhoods of a $k$-tuple $\boldsymbol{i} \in [n]^k$:

$$N_j(\boldsymbol{i}) = \{(i_1, ..., i_{j-1}, i', i_{j+1}, ..., i_k)|i' \in [n]\} \tag{60}$$

$$N_j^F(\boldsymbol{i}) = ((j, i_2, ..., i_k), (i_1, j, ..., i_k), ..., (i_1, ..., i_{k-1}, j)) \tag{61}$$

$N_j(\boldsymbol{i})$ denotes the $j$-th neighborhood of the tuple $\boldsymbol{i}$ used by the WL algorithm, where $j \in [k]$. $N_j^F(\boldsymbol{i})$ represents the $j$-th neighborhood of the tuple $\boldsymbol{i}$ used by the FWL algorithm, where $j \in [n]$. Note that $N_j(\boldsymbol{i})$ is a set of $n$ $k$-tuples, while $N_j^F(\boldsymbol{i})$ is an ordered set of $k$ $k$-tuples. Therefore, we can obtain detailed algorithms of $k$-WL and $k$-FWL as shown in Algorithm 1 and 2.

---

**Algorithm 1:** The $k$-dimensional Weisfeiler-Lehman Algorithm

**Input:** Graph $G = (V, E)$ and the number of iterations $T$
**Output:** The coloring of all $k$-tuples $\mathbf{C}$
1 **Initialization:** The initial coloring $\mathbf{C}^0$ is defined using the isomorphism type of each $k$-tuple
2 **for** $t \leftarrow 1$ **to** $T$ **do**
3     **for** *each $k$-tuple $\boldsymbol{i}$* **do**
4         $\mathbf{C}_{\boldsymbol{i}}^t := \text{HASH}(\mathbf{C}_{\boldsymbol{i}}^{t-1}, (\{\{\mathbf{C}_{\boldsymbol{j}}^{t-1}|\boldsymbol{j} \in N_j(\boldsymbol{i})\}|j \in [k]\}))$
5     **end**
6 **end**
**Result:** $\mathbf{C}$

---

**Algorithm 2:** The $k$-dimensional Folklore Weisfeiler-Lehman Algorithm

**Input:** Graph $G = (V, E)$ and the number of iterations $T$
**Output:** The coloring of all $k$-tuples $\mathbf{C}$
1 **Initialization:** The initial coloring $\mathbf{C}^0$ is defined using the isomorphism type of each $k$-tuple **for** $t \leftarrow 1$ **to** $T$ **do**
2     **for** *each $k$-tuple $\boldsymbol{i}$* **do**
3         $\mathbf{C}_{\boldsymbol{i}}^t := \text{HASH}(\mathbf{C}_{\boldsymbol{i}}^{t-1}, \{\{(\mathbf{C}_{\boldsymbol{j}}^{t-1}|\boldsymbol{j} \in N_j^F(\boldsymbol{i}))|j \in [n]\}\})$
4     **end**
5 **end**
**Result:** $\mathbf{C}$

---

### D.2    THE EXPRESSIVE POWER OF RAYE-WL

We start by introducing WL analogue (RayE-WL) for RayE-Sub to support our next study of the expressiveness of our. Due to limited pages, we only present the core step of RayE-WL: color refinement algorithm, detailed algorithm is provided in Algorithm 3. On subgraph $G_S^i \in G_S$, the color of node $v \in G_S^i$ is refined according to the rule:

$$c_{v,G_S^i}^{t+1} := \text{HASH}(c_{v,G_S^i}^t, N_{v,G_S^i}^t, c_{G_S^i}^{t+1}) \tag{62}$$

$$c_{G_S^i}^{t+1} := \text{HASH}(c_{G_S^i}^t, M_{G_S^i,G}^t, c_G^{WL,t}) \tag{63}$$

$N_{v,G_S^i}^t$ denotes the multiset of colors in $v$'s neighborhood over subgraph $G_S^i$ after the t-th iteration. $c_{G_S^i}^t$ represents the color of the subgraph $G_S^i$ in which node $v$ is existed after the t-th iteration. $M_{G_S^i,G}^t$ denotes the color multiset of all subgraphs of the graph $G$ independently mapped by 1-WL after the t-th iteration, $M_{G_S^i,G}^t = \{c_S^{WL,t}|S \in G_{G_S^i}\}$. $c_G^t$ represents the color of the graph $G$ mapped by 1-WL after the t-th iteration.

We then explore the expressive power of RayE-WL from two aspects: comparing its ability with 1-WL, and studying the expressivity upper bound of RayE-WL.

**Theorems 4.** *(RayE-WL is more powerful than 1-WL.) in distinguishing between non-isomorphic graphs, which is upper bounded by 3-WL.*

*Proof.* We first prove that RayE-WL is strictly more powerful than 1-WL. Given two non-isomorphic graphs $G$, $H$, we first prove that RayE-WL is stronger than 1-WL by two terms. (i) If they can be distinguished by the 1-WL graph isomorphism test, RayE-WL can strictly distinguish them. (ii) There exist graphs that cannot be distinguished by 1-WL but can be distinguished by RayE-WL.

For the first term, $G$ and $H$ can be distinguished by the 1-WL graph isomorphism test, which means $c_G \neq c_H$. We can easily deduce $\{c_{G_S^i}|G_S^i \in G_S\} \neq \{c_{H_S^i}|H_S^i \in H_S\}$, where $G_S$ and $H_S$ are the subgraph sets extracted by $G$ and $H$ respectively. The final the color multiset of $G$ and $H$ is also distinguishable.

---

**Algorithm 3:** The RayE-WL Algorithm

---
**Input:** Graph $G = (V, E)$ and the number of iterations $T$
**Output:** The coloring of all nodes $c^T$
1 **Initialization:** Initialize the color of each node $c^0$, and the subgraph set $G_S$ extracted from $G$
2 **for** $t \leftarrow 1$ **to** $T$ **do**
3     **for** *each subgraph $S$ in $G_S$* **do**
4         **for** *each node $v$ in $S$* **do**
5             $c_S^t := \text{HASH}(c_S^{t-1}, M_{S,G}^{t-1}, c_G^{WL,t-1})$
6             $c_{v,S}^t := \text{HASH}(c_{v,S}^{t-1}, N_{v,S}^{t-1}, c_S^t)$
7         **end**
8     **end**
9 **end**
**Result:** $c^T$

---

For the second term, $G$ and $H$ cannot be distinguished by the 1-WL graph isomorphism test, which means $c_G = c_H$. After $G$ and $H$ are partitioned into subgraphs $G_S$ and $H_S$, there are two existing cases that should be discussed. One is that each subgraph can be distinguished which means $\{M_{G_S^i,G}|G_S^i \in G_S\} \neq \{M_{H_S^i,H}|H_S^i \in H_S\}$. In other words, the independent mapping results based on 1-WL in each subgraph can distinguish between $G_S$ and $H_S$. Thus, we can similarly deduce $\{c_{G_S^i}|G_S^i \in G_S\} \neq \{c_{H_S^i}|S_S^i \in H_S\}$. The final color of each node $c_{v,S}$ is also distinguishable. More importantly, another case is each subgraph still cannot be distinguished which means $\{M_{G_S^i,G}|G_S^i \in G_S\} = \{M_{H_S^i,H}|H_S^i \in H_S\}$. The bottom panel of Figure 1 intuitively describes this case. According to the comparison between their color refinement algorithm, we can observe our RayE-WL is a subclass of 3-WL. Thus, we can conclude that RayE-WL is upper bounded by 3-WL

## E  TIME COMPLEXITY ANALYSIS

In this section, we analyze the time complexity from theoretical analysis and experimental comparison. For each graph $G = (A, X)$, our method has two parts of calculation: subgraphs partition and subgraph-level equivariant learning.

**Time Complexity.** The running time of the subgraphs partition stage main comes from calculating edge weight ($w_{ab}$) and resistance distance ($RD_{ab}$). It is obvious that the process of reweighting edge costs $\mathcal{O}(|E|)$ time. We are only concerned with the resistance distance between nodes connected by edges, thus we do not need to follow the inefficient calculation method with $\mathcal{O}(|V|^3)$. Inspired by Equation 45, we propose a resistance distance approximation method with $\mathcal{O}(|E|)$ time complexity. Specifically, we first precalculate the resistance distance among all the nodes $RD^*$ in the case that all edges have a weight of 1. Then, with the weight $w_{ab}$ of edge $(a, b)$, we approximate the resistance distance $RD_{ab} = RD_{ab}^*/w_{ab}$. This proposal realizes the calculation of resistance distance in $\mathcal{O}(|E|)$ time complexity. Note that this approximate method can only be used to calculate the resistance distance between nodes connected by edges. The cost time of the subgraph-level equivariant learning stage main stems from query mechanism. Its time complexity is $\mathcal{O}(md^2)$, where $m$ represents the number of subgraphs, $d$ indicates the hidden dimension. Due to $d$ is a constant that we set up, the time complexity of *RayE-Sub* is $\mathcal{O}(|E|)$.

**Execution Time.** We compare the execution time (epoch/s) of four baselines (GIN, PNA, GNN-AK and GSAT) on three datasets (BBBP, MUTAG and BA-2Motifs) as shown in Table 3. Note that all experiments are conducted on a Tesla V100-PCIE-16GB GPU, and the backbones of all models are 2-layer GIN, same as *RayE-Sub*. The result shows that

Table 3: The execution time of four baselines on three datasets.

|  | BBBP | MUTAG | BA-2Motifs |
|---|---|---|---|
| GIN | 0.79 | 1.07 | 0.64 |
| PNA | 1.30 | 2.62 | 0.91 |
| GNN-AK | 2.76 | 4.32 | 1.23 |
| GSAT | 3.88 | 6.49 | 2.98 |
| RayE-Sub | 4.06 | 9.86 | 2.65 |

the running efficiency of *RayE-Sub* is competitive, and it achieves interpretability and performance improvements within an acceptable time consumption.

## F SUPPLEMENTARY EXPERIMENTS

### F.1 DETAILS OF THE DATASETS

**BA-2Motifs** is a synthetic dataset created by (Luo et al., 2020) with two graph classes. House motifs and cycle motifs give class labels and thus are regarded as ground-truth explanations for the two classes respectively.

**Spurious-Motif** is a synthetic dataset proposed by (Wu et al., 2022) with three graph classes. Each graph is composed of one base $S$ and one motif $C$. The motif $C$ directly determines the label of the graph. We can create Spurious-Motif datasets with different spurious correlation, which represents the degree ($b$) between the base $S$ and the label. In our implementation, we choose $b = 0.5, 0.7$ and $0.9$ to obtain datasets.

**MUTAG** Debnath et al. (1991) is a binary dataset of molecular property, where nodes are atoms and edges are chemical bonds. Each graph is associated with a binary label based on its mutagenic effect.

**Open Graph Benchmark (OGB)** Hu et al. (2020) is a series of real, large-scale and diverse datasets which is utilized for machine learning on graphs. It covers almost all real-world tasks, including node-level, link-level and graph-level property prediction. We choose MOLHIV, BBBP and SIDER to verify our method.

### F.2 DETAILS OF THE BASELINES

Our baselines are three-fold, including GNN backbones, node-based subgraph learning methods and partition-based subgraph learning models. The details are as follows.

**Backbone baselines.** GCN Kipf & Welling (2016), Graph-SAGE Hamilton et al. (2017) and GIN Xu et al. (2018) are very popular backbones in most practices. These classic MPNNs are limited in their expressive ability, they are still selected as baselines by many subgraph learning methods Zhang et al. (2023); Zhao et al. (2021); Yang et al. (2022); Frasca et al. (2022).

**Node-based subgraph learning methods.** ESAN (Bevilacqua et al., 2022) implements an subgraph equivariant learning architecture and achieve better expressiveness by per-layer aggregation across

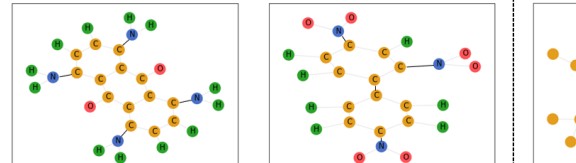

Figure 6: Visualizing the boundary discovered by RayE-Sub, where the **bold connections** are the edges of boundary. **Left panel:** the boundary obtained by RayE-Sub for MUTAG. **Right panel:** the boundary discovered by RayE-Sub for BA-2Motifs.

subgraphs. GNN-AK Zhao et al. (2021) follows a similar manner to develop Subgraph GNNs by considering the star-pattern as the pre-defined substructure. SUN (Frasca et al., 2022) profoundly studies the characteristics of node-based subgraph learning. Further, SUN aligns the permutation group of nodes and subgraphs, and models the symmetry with a smaller single permutation group.

**Partition-based subgraph learning models.** IB-subgraph Yu et al. (2020) first implement the information bottleneck theory in practice on graph learning, which is not only a subgraph learning method based on partition (edge drop) but also an important exploration of interpretability. GSAT (Miao et al., 2022) follows this practice and designs a subgraph extraction strategy with edge deletions based on stochastic attention mechanism. DIR (Wu et al., 2022) splits the input graph into causal and non-causal subgraphs, and utilizes invariant features to construct intrinsically interpretable GNNs.

## G  INTERPRETATION VISUALIZATION

We provide visualizations of the boundary discovered by *RayE-Sub* on two datasets (MUTAG and BA-2Motifs) as shown in Figure 6. According to the results of visualization, we explore (i) whether our method can partition original graph into label-relevant subgraphs, (ii) whether the setup of $\beta = 0.05$ can match the real scenario of the datasets.

Following (Miao et al., 2022), $-NO_2$ and $-NH_2$ in MUTAG dataset are labeled as ground-truth explanations. In our practice, we observe that our *RayE-Sub* always partition $-NO_2$ or $-NH_2$ into a subgraph as shown in Figure 6. We have to acknowledge that our method also splits at label-relevant edges. Excitingly, our partition tends to divide the molecule into a bag of functional groups, which indicates that our method is suitable for real-world tasks. Besides, we find that the number of label-relevant subgraphs is small, only a few edge partition can satisfy the expectation of prediction. Therefore, the setup of $\beta = 0.05$ is appropriate for real-world tasks.

Following (Luo et al., 2020), house motifs and cycle motifs give class labels and thus are regarded as ground-truth explanations for the two classes respectively. Specifically, each base graph is generated using the BA model and will be attached with two house motifs or three house motifs randomly. The number of house motifs represents the graph class. The primary goal of this task is to identify house motifs. We observe that our *RayE-Sub* always partition house motif (pink nodes) into a subgraph. Similar to MUTAG, BA-2Motifs also needs only a few number of boundary edges to achieve accurate prediction.

In conclusion, our *RayE-Sub* can partition original graph into label-relevant subgraphs, and the setup of $\beta = 0.05$ is appropriate for the tasks of these datasets.

## H  FUTURE WORKS

We can still improve our work from following two aspects. In the subgraphs partition stage, how to set an personalized and optimal partition rate $\beta$ via domain knowledge across different datasets is still unexplored. And in the subgraph aggregation stage, it is interesting to investigate more powerful equivariant architectures for ordered aggregations.

