# OpenReview forum: "RayE-Sub: Countering Subgraph Degradation via Perfect Reconstruction"
_ICLR.cc/2024/Conference — ICLR 2024 Conference Withdrawn Submission_

### Official Review · Reviewer_7R4q · 2023-10-31

**Soundness:** 2 fair
**Presentation:** 1 poor
**Contribution:** 2 fair
**Rating:** 3
**Confidence:** 4

**Summary:**

This paper studied the problem of subgraph learning. The paper delves into the limitations of current Message Passing Neural Networks (MPNNs) in graph representation and explores the potential of subgraph learning as a predominant solution. The authors critically examine existing subgraph learning strategies, highlighting their limitations, particularly in terms of reconstruction ability. The authors propose a new method – Rayleigh-resistance Equivariant Subgraph learning architecture (RayE-Sub) – that ensures perfect reconstruction of extracted subgraphs. The experimental results on several datasets demonstrate the effectiveness of the proposed method.

**Strengths:**

1. The proposed method works reasonably well.  Empirical results show that proposed method can achieve better performance compared to baselines
2. The paper offers both theoretical proofs and empirical evaluations.

**Weaknesses:**

1. The notion of "perfect reconstruction" is not well-explained. It would be beneficial if the authors clearly defined what constitutes "perfect" in their context. Moreover, why can "perfect reconstruction" promote the generalization of subgraph learning?
RayE-Sub appears to be complex. Practical implementation and scalability might be concerns.

2. The writing is very poor. Many terms, such as "equivariance," "permutation," "high-frequency," and "homogenization," are not well-explained and defined in the context of the paper.

3. This paper does not make a clear connection with the presented theoretical results and contextualize this work in the existing literature. It is difficult to discern what's new and what's already part of the existing literature. Additionally, the theoretical analyses could have been written more clearly. They are difficult to understand currently. Consider adding a symbol table to the paper. The presentation of the proposed method is too brief and intertwined with theoretical analysis, making the method challenging to read and understand.

4. It's also unclear whether the paper makes a sufficient experimental contribution. Each theorem, while interesting, is derived fairly straightforwardly from existing results. The experiments are conducted over multiple datasets and compared against numerous baselines, but the results for the graphs displaying spurious correlation are unexplained and concerning. Can the authors provide a better explanation for the performance in the graphs with spurious correlation?

5. Some experimental results might provide a better understanding. For instance, evaluating the performance of other competitive methods on graphs with spurious correlation would be informative.

**Questions:**

Please see the "Weaknesses" section.

---

### Official Review · Reviewer_h1tc · 2023-10-31

**Soundness:** 3 good
**Presentation:** 3 good
**Contribution:** 3 good
**Rating:** 6
**Confidence:** 3

**Summary:**

Subgraph-based representation learning has received increasing attention in recent years for its inherent interpretability and improved expressive power. However, recent approaches face degraded subgraph extraction, which limits their performances. This work attributes this limitation to the imperfect reconstruction of original graph information, and proposes the RayE-Sub framework with Rayleigh-quotient-based subgraph extraction and attention-based equivariant subgraph aggregation. Empirical evaluation and theoretical justification validate the effectiveness of the proposed method.

**Strengths:**

This work is well-presented and easy to follow. The proposed RayE-Sub method is novel and enjoys good graph spectral implications with promising subgraph extraction performances. Empirical performance and theoretical justification validate the effectiveness of the proposed method.

**Weaknesses:**

Yet, the reviewer has several concerns about clarity on some technical details and related works.

1. Partition-based subgraph discovery policies, such as GIB [1] and GSAT [2] extract label-relevant subgraphs for graph classification. Since they extract one subgraph for an input graph instead of a bag of subgraphs, what does the 'order information of Partition-based subgraph discovery policies' refer to in Line 8, Page 2?

2. Prior work on equivariant subgraph network [3] usually maintain equivariance to subgraph ordering when aggregating subgraph representations to obtain graph representation. How does the SIAMESE-QUERY NETWORK maintain such equivariance?

3. Baseline: IB-Subgraph [1] -> GIB [1]. Also, it is recommended to discuss this work in Related Work since GIB is one of the first to generalize the IB principle to subgraph learning and inspire several follow-up works.

4. The last line of Page 5: existence->exist.







[1]. Graph Information Bottleneck for Subgraph Recognition. ICLR 2021.
[2]. Interpretable and Generalizable Graph Learning via Stochastic Attention Mechanism. ICML 2022.
[3]. EQUIVARIANT SUBGRAPH AGGREGATION NETWORKS. ICLR 2022.

**Questions:**

The authors are encouraged to address the concerns in Weaknesses.

---

### Official Review · Reviewer_NoaS · 2023-11-01

**Soundness:** 1 poor
**Presentation:** 1 poor
**Contribution:** 2 fair
**Rating:** 3
**Confidence:** 4

**Summary:**

This paper proposes RayE-Sub, an approach for improving subgraph-learning based MPNNs by improved subgraph selection, through the use of extracting non-overlapping "perfect" subgraphs which enable reconstruction of the input graph without redundancy.  The authors evaluate their approach on a number of graph classification tasks showing improvements over baseline subgraph-learning methods.

**Strengths:**

S1. The authors study an interesting problem, focused on improving subgraph selection for subgraph-learning based MPNNs.  Subgraph selection is indeed a heuristic for such methods, so developing better understanding here is valuable.

S2. The authors support their method with theoretical analysis about expressiveness.

**Weaknesses:**

W1. The argument about redundancy and problems it creates for subgraph learning are not very strongly motivated -- much of the paper seems to rely on this motivation.

W2.  A number of the theorems and mathematical statements seem imprecise/vague or misleading -- I raise several of these issues below.

W3. The results seem to not be convincingly better than alternatives, often well within the margin of error compared to other subgraph-based methods which do not rely on the "perfect reconstruction" argument the authors propose.

**Questions:**

- "Both of them have revealed effectiveness in practices, where the former is with more simplicity, while the latter is more interpretable." (p1) -- Is this true?  I'm not sure what is more interpretable about partition-based solutions.  Subgraph-based solutions which center around individual nodes are usually interpreted as predictions on a node's local graph structure.  The authors might want to clarify here.

- The introduction mentions that reconstruction of the original graph without redundancy is a property desirable in subgraph-learning approaches.  It is not well-explained why this is the case.  Some intuition would be very helpful.

- What are $m$ and $k$ in Figure 1?

- There is a relevant related work the authors might wish to consider in the line of expressive GNNs outside the k-WL test hierarchy [1].

- Using subscript notation for $i$ for $x_i$ is confusing since $i$ is used as a superscript to index the $i$th subgraph in multiple other variables.

- Lemma 1 seems to ignore nonlinearities (assuming matrix operators for updation and aggregation) -- could you clarify this explicitly?

- Not sure I understand this: "If extracted subgraphs fail to limit previous MPNN’ aggregation trend, they will not bring better ability of distinguishing" (under Lemma 1).

- How does Theorem 1 use the ground-truth aggregation matrix $M$?  This seems unreferenced in the theorem/proof.  Also it is not clear that the aggregation matrix of the whole graph is the sum of the aggregation matrices of the individual graphs.  e.g. if we have an (undirected) line-graph with nodes A-B-C (all edges with weight 1), then node A has an aggregation matrix reflecting an edge from A-B, node B also has an aggregation matrix reflecting the same edge; the resulting aggregation matrix of the whole graph has only edge weight 1.  Maybe I'm misunderstanding.

- The logical jump from Theorem 1 to the conclusion that subgraph degradation exists isn't clear.  This could benefit from some careful phrasing or maybe a toy example to illustrate carefully.

- Regarding Sec 3.2, "Obviously, this design is not appropriate for real-world tasks of subgraph learning. " -- this doesn't follow to me.  Doesn't the design of invariance or equivariance strongly depend on the application of the real-world task? Can you clarify why equivariance is preferred to invariance in the real-world tasks considered in this paper?  There are many tasks in which equivariance is not desirable.

- 3.3's discussion of "Node-based subgraphs with redundant information." still leaves questions as to why redundancy is bad.  What does a 1-WL-similar message passing space mean?  Some empirical observations demonstrating that increased redundancy in node-based subgraph learning results in worsened performance in practical tasks would help motivate this point.

- Theorem 2 tries to relate the Resistance Distance (RD) to the Rayleigh Quotient (RQ), but RD does not seem to use any terms reflecting node features while RQ does -- this is confusing.  How are the two terms related?  I am not sure if there are derivations in the appendix showing this, but with the main paper content... one reflects features and the other does not.

- The discussion under Eq7 seems to all be concerned with how to extract a bag of non-overlapping subgraphs from the input graph.  There is not much intuition for me (as a reader) as to why the proposed approach to optimize RD/RQ is the preferred way to do this.  Any number of combinatorial choices exist for extracting non-overlapping subgraphs from the input graph.  What is special about this choice?

- In many cases, your bolded results are well-within the error-margins of other approaches and their standard deviations.  Are the improvements truly significant (e.g. by a two-sample t-test?)

[1] A Practical, Progressively-Expressive GNN (NeurIPS'22)